# Autophagic-Related Proteins in Brain Gliomas: Role, Mechanisms, and Targeting Agents

**DOI:** 10.3390/cancers15092622

**Published:** 2023-05-05

**Authors:** Cristina Pizzimenti, Vincenzo Fiorentino, Mariausilia Franchina, Maurizio Martini, Giuseppe Giuffrè, Maria Lentini, Nicola Silvestris, Martina Di Pietro, Guido Fadda, Giovanni Tuccari, Antonio Ieni

**Affiliations:** 1Translational Molecular Medicine and Surgery, Department of Biomedical and Dental Sciences and Morphofunctional Imaging, University of Messina, 98125 Messina, Italy; 2Department of Human Pathology in Adult and Developmental Age “Gaetano Barresi”, Pathology Section, University of Messina, 98125 Messina, Italy; 3Department of Human Pathology in Adult and Developmental Age “Gaetano Barresi”, Oncology Section, University of Messina, 98125 Messina, Italy

**Keywords:** autophagy, autophagy-related proteins, gliomas, glioblastomas, prognosis, treatment

## Abstract

**Simple Summary:**

The aim of the present review is to discuss the autophagy, a well-known cellular process, able to remove damaged intracellular organelles as well as macromolecules and misfolded proteins. A dual role, as tumour promoter and tumour suppressor, has been attributed to autophagy. Therefore, we would analyse molecular mechanisms and regulatory pathways of autophagy, mainly concerning human astrocytic neoplasms. Moreover, information about relationships between autophagy, the tumour immune microenvironment, and glioma stem cells are furtherly illustrated. Drugs with higher selectivity for autophagy are actually developing and hopefully applied in the future to clinical practice. This modern perspective could help in the selection of patients with gliomas that are most likely to respond to the therapy of autophagy–inhibition.

**Abstract:**

The present review focuses on the phenomenon of autophagy, a catabolic cellular process, which allows for the recycling of damaged organelles, macromolecules, and misfolded proteins. The different steps able to activate autophagy start with the formation of the autophagosome, mainly controlled by the action of several autophagy-related proteins. It is remarkable that autophagy may exert a double role as a tumour promoter and a tumour suppressor. Herein, we analyse the molecular mechanisms as well as the regulatory pathways of autophagy, mainly addressing their involvement in human astrocytic neoplasms. Moreover, the relationships between autophagy, the tumour immune microenvironment, and glioma stem cells are discussed. Finally, an excursus concerning autophagy-targeting agents is included in the present review in order to obtain additional information for the better treatment and management of therapy-resistant patients.

## 1. Introduction

Autophagy is a catabolic cellular process that maintains cellular homeostasis through the degradation, elimination, and recycling of damaged substrates, such as organelles, macromolecules, and misfolded proteins [1,2]. Under physiological conditions, autophagy activity is at the basal level, but it can be augmented under stressful conditions, such as cell death, nutrient deprivation, oxidative stress, and pathogen invasion [3,4]. The activation of autophagy starts with the formation of a double-membrane vesicle, the autophagosome, where the substrates’ degradation takes place, and it is controlled by several autophagy-related proteins (ATGs), such as the UV radiation resistance-associated gene (UVRAG) [5,6,7], phosphatidylinositol 3-kinase catalytic subunit type 3 (PIK3C3) [8], microtubule-associated protein 1 light chain 3 (LC3) [9], Beclin-1 [10,11], activating molecule in Beclin-1 regulated autophagy (AMBRA 1) [12,13], unc 51-like kinase complex (ULK) [14], and the ubiquitin-binding protein (p62) [15,16].

It is interesting to note that autophagy has been demonstrated to have a double role, both as a tumour promoter and as a suppressor [1,17,18]. As a matter of fact, autophagy promotes cancer initiation and survival through the recycling of intracellular substrates in order to sustain tumour metabolism, contributing to the acquisition of resistance to treatments [19,20]. However, autophagy also acts as a tumour suppressor by removing damaged proteins and organelles, in order to protect cells from reactive oxygen species (ROS), inflammation, necrosis, and other causes of genomic instability [21,22].

In this paper, after a review of the autophagic molecular mechanisms, the role of autophagy-targeting agents in human brain gliomas is examined, taking into consideration the current relevant literature regarding astrocytic lineage. Finally, new developments in autophagy target treatments for high-grade glial tumours are examined.

## 2. Molecular Mechanisms of Autophagy

The autophagic process can occur as a consequence of one of three different mechanisms: (1) macroautophagy, in which the degradation of the substrates occurs after inclusion in an autophagosome; (2) microautophagy, in which the substrates are directly invaginated into the lysosomal membrane; and (3) chaperon-mediated autophagy (CMA), in which a heat-shock protein (HSC70) serves as a molecular chaperone for the substrates containing the KFERQ motif and facilitates their translocation into the lysosome through the lysosomal-associated membrane protein 2A (LAMP2A) receptor, promoting their degradation [23,24].

Several autophagy-related genes encoding for kinases, phosphatases, and GTPases are involved in the autophagy process, acting in five different steps: initiation, nucleation, elongation, completion, and fusion with lysosome for demolition [25,26,27,28]. In physiological conditions, the AMP-activated protein kinase (AMPK) controls the activation of the mammalian target of rapamycin (mTOR), leading to the hyperphosphorylation of unc-51-like kinase 1 and 2 (ULK1 and ULK2) and preventing autophagy initiation [29]. By contrast, under stressful conditions, an increase in AMPK inhibits mTOR, which leads to the activation of the ULK1/2 complex, and its association with Atg13, Atg10, and FIP200 causes the relocation of the complex to the membrane of the endoplasmic reticle [14,30,31]. The autophagosome, a double-membrane vesicle, is formed during the nucleation phase [26]. Vesicle nucleation is regulated by Beclin-1 and class-III PI3K (PI3K-III) complexes, such as PI3K3/VPS34 and p150/VPS15 [27,32]. In particular, the key regulator of autophagy, Beclin-1, and its interaction with the UVRAG, AMBRA-1, and Bax-interacting factor-1 (Bif-1) results in the formation of the Beclin-1/PI3K-III complex [6,7,33,34]. This cascade leads to the production of phosphatidylinositol 3-phosphate (PI3P) and to the recruitment of several proteins involved in autophagosome formation and maturation [25,26,27]. The elongation phase is regulated by two ubiquitin-like protein conjugation systems: phosphatidylethanolamine (PE)/microtubule-associated protein 1 (MAP1)/light chain 3(LC3)/Atg8 and Atg5/Atg12/Atg16 [35,36,37]. The ubiquitin-like protein Atg12 binds Atg5 and Atg16, thus forming a complex located on the outer surface of the autophagosome that mediates the binding of LC3 to the autophagosome membrane [38,39]. MAP1/LC3/Atg8 is cleaved by Atg4B into the cytosolic form LC3-I, exposing a reactive glycine residue in the C-terminus end of MAP1-LC3/Atg8 and allowing for binding with PE through Atg7 and Atg3 in order to form LC3-II [40,41]. LC3-II mediates the closure of the membrane of the autophagosome and its fusion with the lysosome; then, it is degraded and released into the cytosol [25,26,42].

The fusion of the autophagosome with the lysosome depends on important proteins, such as the N-ethylmaleimide-sensitive factor attachment protein receptor (SNARE) protein—Syntaxin 17 (Stx17) and RAB7, a membrane tethering homotypic fusion and vacuole protein-sorting (HOPS) complex [43,44]. Stx17 forms a trans-SNARE complex to allow for membrane fusion with an association of Stx17 to the guanosine triphosphatase, named immunity-related GTPase M (IRGM) [45]. Furthermore, an association of this complex with Atg8proteins achieves a so-called autophagosome recognition particle (ARP) needed for autophagosome assembly [45]. Moreover, UVRAG plays a fundamental role in autophagosome and lysosome fusion [43]. In fact, UVRAG recruits the C-vacuolar protein (C-VPS) on the autophagosome and subsequently promotes the activity of the Rab7-GTPase along with the proteins LAMP-1/2, resulting in the fusion of the autophagosome and lysosome [27,46]. The substrates needed for degradation are identified via specific domains, such as the LC3-II interacting regions (LIRs), a PB1 oligomerization domain, and a ubiquitin-associated (UBA) region, that interact with autophagy receptors [25,26,27]. Moreover, p62 or sequestosome-1 (p62/SQSTM1) binds ubiquitinated proteins through UBA domains, forming aggregates that are recognized by LC3-II on the inner surface of the autophagosome [16,47,48,49]. Finally, the substrates inside the newly formed inner membrane are degraded by lysosomal hydrolases in the final step of the autophagic process, while the degraded material is recycled and returned to the cytosol [25,26,27].

## 3. Pathways of Autophagy Regulation

The regulation of autophagy depends on several different signalling pathways [29,50,51,52]. Firstly, the AMPK/mTOR pathway is the most known and analysed [29,50]. In detail, mTOR is composed of two complexes. The first is mTORC1 that regulates cell growth, energetic metabolism, and autophagy. It is sensitive to rapamycin, while mTORC2, which regulates cell proliferation and cytoskeleton organization, is not sensitive to rapamycin [29,50,53]. In a nutrient-rich environment, AMPK is inactive while mTORC1 is active, thus inhibiting autophagy through the phosphorylation of Atg13, ULK, and AMBRA [14,29,54]. The AMPK/mTORC1/ULK1 pathway can also regulate the PIK3C3/VPS34 complex by controlling Beclin-1 and VPS34, depending on the presence of ATG14L [55]. When ATG14L is present, Beclin-1 is activated by the AMPK through phosphorylation, promoting autophagy [27,55,56]. Moreover, the PI3K/AKT/mTOR pathway is involved in tumour cell growth, proliferation, metastatic progression, and angiogenesis, as well as being associated with several disorders, including tumours and neurodegenerative disorders [50,57,58,59]. On the other hand, PI3K, whose activation depends on the association with different proteins, such as growth factors, is involved in the production of PI3P, leading to the activation of AKT through its phosphorylation and subsequently to the inhibition of autophagy [51,60,61]. Another important control pathway is the MAPK/ERK, involved in a wide range of cellular functions, such as proliferation, differentiation, apoptosis, cellular stress control, and autophagy regulation [52,62]. Under stressful conditions, the activation of the MAPK/ERK by the AMPK promotes the disassembly of the mTOR complex and its inhibition with a significant increase in Beclin-1 activity and the start of the autophagic process [63]. High Beclin-1 levels lead to cytodestructive autophagy compared to moderate Beclin-1 levels, which, on the contrary, induce cytoprotective autophagy [63].

Additional transcription factors have been involved in the regulation of autophagy; in particular, mTOR and ERK2 control the phosphorylation of the transcription factor EB (TFEB) usually located in the cytoplasm [64,65]. When TFEB is dephosphorylated, it is translocated to the nucleus, and it can activate the expression of several autophagy-related genes, including BECN1, WIPI1, ATG9B, NRBF2, GABARAP, MAP1LC3B, ATG5, SQSTM1, UVRAG, and RAB7, regulating autophagy initiation, autophagosome formation, and fusion [64]. Moreover, the Forkhead box class O (FoxO) family of transcription factors is composed of FoxO1/FKHR, FoxO3/FKHRL1, FoxO4/AFX, and FoxO6, which regulate cellular homeostasis, autophagy, angiogenesis, tumour growth, and metastasis [66,67,68]. FoxO proteins translocate from the cytoplasm to the nucleus and induce the expression of several genes implicated in the autophagic process (ULK, Beclin-1, ATG14, GABARAP, MAP1LC3B, ATG4, TFEB, and Rab7) [66]. FoxO proteins can also increase the expression of Sestrin 3 (Sesn3), which can activate the AMPK and inhibit the mTORC1 [69,70]. It has been shown that cellular functions in a hypoxic environment may be regulated by the transcription factor hypoxia-inducible Factor-1 (HIF-1) [71,72]; HIF-1 is constituted by two subunits, 1α and 1β, able to form HIF-1 when HIF-1α is translocated to the nucleus during hypoxic conditions [71,72]. When the HIF-1 complex is formed, it becomes active and induces the expression of BINP3, BNIP3L, Beclin-1, and Atg5, regulating autophagy, cell proliferation and survival, and angiogenesis [73,74].

TP53, PTEN, STAT3, and NF-κB are also transcription factors involved in the regulation of autophagy [75,76,77,78]; specifically, p53 may operate as a constitutive inhibitor of autophagy when it is localized in the cytoplasm [75]. However stress-inducible systems, such as those that control p53, STAT3, and NF-κB, not only orchestrate delayed autophagic responses as they activate specific genetic programs but also promote the rapid activation of the autophagic machinery [75,76,77,78].

## 4. The Role of Autophagy in Gliomas

Diffuse gliomas are the most common primary brain tumours, classified according to the integration of their histopathological and genetic features [79]. The most recent 2021 WHO classification of CNS tumours highlights the increasing importance of molecular diagnostics in glial tumours considering the impact they have on the classification of these tumours [79]. Specifically, adult-type diffuse gliomas are classified into two groups of IDH-mutant gliomas: (1) astrocytoma IDH-mutant (grade 2 to 4) and (2) oligodendroglioma IDH-mutant, 1p/19q-codeleted (grade 2 to 3), as well as glioblastoma IDH-wildtype (GBM, grade 4) [79]. Low-grade IDH-mutant gliomas (LGGs) are low-cell-density, diffusely infiltrating, and slow-growing tumours, composed of well-differentiated glial (astrocytic or oligodendroglial) cells, with mild nuclear atypia and a lack of mitosis, necrosis, or microvascular proliferation [79,80]. IDH-mutant gliomas are genetically defined by the presence of the IDH1 or IDH2 gene mutation [81]. They are associated with a younger age and longer survival [82,83,84]. The increase in the tumour grade, histologically characterized by the presence of severe nuclear atypia, necrosis, and microvascular proliferation, is accompanied by the accumulation of several genetic alterations, such as the loss of the function of protein 53 (TP53) and ATRX, TERT promoter mutation, or homozygous deletion of CDKN2A/B, the latter strongly associated with unfavourable prognosis [83,85]. IDH-wildtype glioblastomas are high-grade, widespread infiltrating gliomas, accounting for 45–50% of all primary malignant brain tumours; they preferentially affect older adults in the 55–85 year range and are characterized by rapid progression and poor prognosis [79]. Histologically, GBMs are high cellular tumours composed of atypical glial cells with marked pleomorphism; the diagnostic features include brisk mitotic activity, microvascular proliferation, and necrosis, with or without palisading [79,80]. GBMs are also characterized by a wide range of genetic alterations, including the amplification and rearrangement of EGFR, TERT promoter mutations, the gain of chromosome 7 and loss of chromosome 10 (+7/−10), and TP53 and PTEN mutations [82,86,87,88].

Due to their intratumoral heterogeneity, high-grade gliomas (HGGs) are refractory to surgical treatment, radio-chemotherapy, and immunotherapy, and the overall survival is 15–18 months, despite the treatment [79,89]. In particular, the overactivation of the tyrosine kinase receptors, such as the epidermal growth factor receptor (EGFR), the platelet-derived growth factor receptor (PDGFR), and the vascular endothelial growth factor receptor (VEGFR), is responsible for the tumour progression and the resistance to therapy of high-grade gliomas [90,91,92]. These proteins induce the activation of genetic signalling pathways that control cell proliferation and migration, angiogenesis, apoptosis, and autophagy, such as RAS/RAF/MPAK and PI3K/AKT/mTOR [27,28].

Autophagy has been known to have a dual role in promoting or suppressing tumour initiation and growth in different types of cancers, including gliomas [1,17,18]. Specifically, autophagy may act as a tumour promoter by recycling substrates for sustaining tumour metabolism and neoplastic survival under adverse circumstances, such as hypoxic stress or nutrient deprivation [19,20]. Alternatively, autophagy shows a role as a tumour suppressor and inhibitory function by removing damaged substrates and organelles, protecting cells from ROS, inflammation, necrosis, and other causes of genomic instability [21,22,23].

Recently, some reports have documented hyper-activation of CMA in GBM through the expression of LC3B, LAMP1, and LAMP2A, with their downregulation due to curcumin [24,25,26]; however, the real implication of CMA-mediated degradation in GBM is still debated. With reference to selective types of autophagy in gliomas (e.g., mitophagy, ER-phagy, lysophagy, etc.), some studies have documented an inhibition of mitophagy, partially reverted cannabidiol-induced glioma cell death, hypothesizing the favourable role of mitophagy [21]. However, the induction of mitophagy by FOXO3a may protect gliomas from TMZ-induced cytotoxicity [21,27]; it has been suggested that early mitochondrial dysfunction and HMOX1 activation may synergize to trigger lethal mitophagy, contributing to the cell death of natural compound AT 101 in glioma cells [93]. In addition, ER-phagy is essential for the proliferation and clonogenicity of mutant IDH1 gliomas due to the downregulation of phospholipid biosynthesis [21,27]. Moreover, autophagic cell death may be triggered by loperamide (LOP) in glioblastoma cells [94]. In detail, LOP may also induce an engulfment of large ER fragments within autophagosomes and lysosomes, as documented in morphological microscopic investigations [94]. LOP-induced reticulophagy and cell death are predominantly mediated through the reticulophagy receptor RETREG1/FAM134B and, to a lesser extent, TEX264, confirming that ER-phagy receptors can promote autophagic cell death [94]. Finally, lysophagy, selective autophagy for damaged lysosomes, has been considered to be a promising therapeutic target for GBM [21,27].

### 4.1. Autophagy as a Tumour Suppressor in Gliomas

Several studies have shown how autophagy acts as a tumour suppressor in gliomas and how its decreased activity is associated with HGGs compared to LGGs that show a more sustained autophagic activity [95,96,97,98,99] (Figure 1). As already mentioned, high AKT levels and mTOR activation are associated with an inhibition of autophagy initiation and are strongly associated with high-grade gliomas compared to low-grade tumours [95,100,101]. Lower Beclin-1 and LC3-II expression have been reported in GBMs compared to LGGs [97,98]; conversely, a high expression of Beclin and LC3 is correlated with a better survival in GBM patients [97,98]. In particular, it has been suggested that higher levels of Beclin-1 can induce apoptosis through binding with Bcl-2 and Bcl-xL and, subsequently, the activation of the proapoptotic proteins Bax and Bak [102,103,104]. Beclin-1 is regulated by EGFR, whose overexpression decreases Beclin-1 levels and promotes tumour progression [105]. Shukla et al. reported that the deletion of fundamental autophagy-related genes for autophagosome initiation and elongation, such as Beclin-1, UVRAG, BIF-1, FIP200, Atg4, and ATg5, as well as a lower expression of ULK1/2, favoured the malignant transformation of astrocytic cells [96]. Moreover, autophagy acts as a tumour suppressor by removing p62 aggregates, whose accumulation causes oxidative stress and tumour initiation, proliferation, and migration [99,106]. Furthermore, Xu at el. reported that a lower expression of p62/SQSM1 significantly decreased ERK phosphorylation, attenuating the proliferation and invasion of glioma cells induced by Guanylate binding proteins-3 (GBP) in vitro [107]. Autophagy can also modulate tumour suppression through the induction of apoptosis operated by Atg5 and its association with proapoptotic proteins [104]. microRNA (miR) gene expression has been reported to regulate autophagic activity. miR-33a and miR-224-3p overexpression inhibits the tumour suppressor UVRAG [108] and the ATG proteins Atg5 and FIP200 [109], respectively, and correlates with poor prognosis in glioblastomas. The expression of Beclin-1 is inhibited by miR-34-5p and miR-5195-3p, favouring migration, invasion, and apoptosis in neoplastic cells [110]. Autophagy limits tumour growth by inducing cellular senescence [111,112]. On the other hand, Temozolomide (TMZ) acts as an autophagy inducer, resulting in senescence in glioma cells [113,114]. In addition, Resveratrol has been reported to improve TMZ toxicity by increasing ROS production and inducing senescence [115,116]. Similarly to flavokawai nB (chalcone), it is able to arrest cellular proliferation in U87, T98, and U251 GBM cell lines with subsequent senescence [117]. It has been reported that adenovirus strains expressing single shRNA specific to c-Met (shMet) induce an increase in the Beclin-1 and LC3-II levels and the inhibition of the AKT/mTOR pathways, promoting autophagy and senescence.

### 4.2. Autophagy as a Tumour Promoter in Gliomas

Several studies have demonstrated how autophagy can act as a tumour promoter in gliomas through the induction of progression and recurrence and the resistance to treatment [118,119,120,121,122,123] (Figure 1).

It has been suggested that one of the mechanisms able to determine brain tumour progression may be represented by hypoxia, determining the activation of the hypoxia-inducible factor 1-alpha (HIF-1α), which induces autophagy through the transcription regulation of autophagic genes [74,124,125]. In addition, HIF-1α also stimulates angiogenesis to have more oxygen and nutrients available for the survival of neoplastic cells via VEGF upregulation [124,125]. However, the rate of hypoxia and the expression of angiogenic factors may be directly related to neoplastic grade and, consequently, to a worse prognosis in brain human gliomas.

It has been reported that the suppression of ULK1, Atg7, and Atg13 favours a reduction in tumour growth [118]. It is also known that the overexpression of LC3 and p62 is correlated with poor prognosis in high-grade gliomas [119], as well as an overexpression of ULK1/2 and TFEB [120]. Additionally, high levels of p62 and Dram1 have been reported to induce cell migration in GBMs and to be associated with poor prognosis in these tumours [121]. The overexpression of LC3 and Beclin-1 is also associated with shorter survival in low- and high-grade gliomas [122]. High Atg4c levels have also been observed in gliomas, while the decrease in this protein is associated with apoptosis, autophagy inhibition, and a greater sensitivity to TMZ [123]. However, some studies have focused on the p62 level in different glial neoplastic samples [119,126,127]; in detail, an increase in p62 expression has been progressively detected from low- to high-grade gliomas with prognostic value, although no correlation with isocitrate dehydrogenase (IDH) mutation status has been documented [119,126,127]. Therefore, it could be argued that p62 overexpression stimulates the classical autophagic pathway, allowing for GBM cell survival by antagonizing apoptosis and producing drug resistance to proteasome inhibitors [128,129]. Alternatively, an accumulation of the autophagy substrate p62 may reveal a defective process that cannot degrade its substrates [126]. Therefore, p62 may act as a tumour promoter in glioma cells, not only by regulating autophagy but also by interfering with proliferation, migration, and TMZ resistance [130]. In addition, the activation of HIF-1α under hypoxic conditions is associated with tumour grade and poor prognosis in HGGs [125,131]. HIF-1α promotes autophagy induction and an increase in Beclin-1, Atg5, and BNIP3L, as well as angiogenesis, through the regulation of VEGF [74,124]. Several studies have demonstrated that the diminished expression of VEGF and HIF-1α in U87 glioma cells led to a reduction in the vasculogenic mimicry (VM) lesions, whose overexpression correlates with tumour grade and poor prognosis in glioblastomas, together with a higher expression of Atg5 and pKDR/VEGFR-2, the latter also inducing the activation of the PIK3/AKT pathway in gliomas [132,133]. Furthermore, hypoxia has been reported to induce the degradation of caveolin-1 (Cav-1) through the activation of pro-autophagy factors, such as BNIP3L, LC3, BNIP3, ATG16L, HIF-1α, and NF-κB. Under physiological conditions, Cav-1 suppresses autophagy by binding and inactivating ATG5, ATG12, and LC3B [134] (Figure 2). A decrease in Cav-1 is associated with tumour cell growth but also with a high expression of monocarboxylate transporters, such as MCT4 and MCT1 (two promoters of tumour growth and progression under hypoxic conditions) [135,136,137] (Figure 2). Autophagy inhibits anoikis (programmed cell death in cells upon detachment from the extracellular matrix), favouring tumour cell invasion and metastatisation [138,139] (Figure 2). Autophagic genes, including Atg5, Atg7, and ULK, are overexpressed in detached cells from glioblastoma, thus preventing anoikis and promoting tumour growth [140,141] (Figure 2). Under metabolic stress, autophagy promotes tumour cell survival by inducing cellular dormancy, a temporary state of arrest of cellular growth that lingers until the end of the stress cause [142]. HIF-1α is one of the proteins responsible for cellular dormancy, and it is linked to poor survival in GBM patients who receive TMZ [142]. Moreover, a prolonged TMZ administration can induce dormancy in glioma cells [142,143]. Malat1 (a long noncoding RNA) promotes cell proliferation by inhibiting miR-101, which downregulates the expression of autophagy-related genes, such as STMN1, RAB5A, and ATG4D; the overexpression of Malat1 is significantly increased in GBM compared to the adjacent normal tissue [144].

It is well known that explanations for the controversies of whether the autophagy pathway promotes survival or death are still debated. In fact, the balance between pro-survival or pro-death autophagic factors may be strongly related to their relationships, since low to moderate levels of autophagy activation become cytoprotective, while high autophagic levels develop cytotoxicity. It has been previously argued that cell lethal excessive autophagy reflects enforced, drug-induced overactivation of autophagy rather than an imbalance of autophagic factors in gliomas [145]. In particular, it has been shown that cannabinoids as well as tricyclic antidepressants (imipramine) and anti-coagulants (ticlopidine) may induce the cell death of cancer cells through autophagic activation, even if non-transformed astrocytes appear resistant to the cannabinoid killing action [145,146].

## 5. Autophagy and the Tumour Immune Microenvironment

The tumour immune microenvironment (TIME) in gliomas is characterized by the presence of tumour-associated macrophages/microglia (TAMs/MG), myeloid-derived suppressor cells (MDSCs), dendritic cells (DCs), neutrophils, and tumour-infiltrating lymphocytes [147,148,149,150]. Intercellular homeostasis and the growth of gliomas are maintained because of the involvement of the TIME cellular constituents, while neoplastic glial elements may be able to recruit immune cells in order to reach immune suppression and evasion [151]. Despite gliomas being known to have a low immunogenic phenotype compared to other tumours [152], some autophagic mechanisms have been reported to modulate immune cells, allowing them to promote an antitumour immune response or, conversely, induce tumour immune tolerance [153]. Autophagy, induced by c-Jun N-terminal Kinase (JNK) activation and the blocking of the Atg5 cleavage, promotes monocytes’ differentiation into macrophages, produces cytokines, and prevents monocyte apoptosis [153,154]. Since macrophages degrade phagocytosed cells through LC3-associated phagocytosis, the inhibition would improve antitumour immunity [155]. Moreover, hypoxia stimulates the release to exosomes containing IL-6 and miR-155-3p from glioma cells that promote autophagic activity in TAMs, such as M2 phenotype polarization through the STAT3 pathway, thus facilitating tumour progression and metastasis [156,157,158]. In addition, it has been demonstrated that M2 macrophage-sourced exosomal miR-15a and miR-92a contribute to inhibiting glioma invasion and migration via the PI3K/AKT/mTOR pathway [159]. However, it has been reported that the inhibition of mTOR favours the M1 phenotype polarization of TAMs, resulting in increased IL-12, decreased IL-10, and reduced tumour angiogenesis [160]. Moreover, a low expression of Atg16L promotes the production of proinflammatory cytokines, such as IL-1β and IL-18, suggesting that autophagy regulates inflammatory activation [161]; finally, Beclin-1 can regulate inflammation through MG in mouse models via NRLP3 [162].

It has been reported that MDSCs and neutrophils can have an immunosuppressive function in glioblastomas [154,163]. Specifically, autophagy inhibition promotes apoptosis in MDSCs and enhances the MHCII expression for tumour-specific CD4+ T cells’ activation, inducing inflammation [164,165,166]. The inhibition of autophagy has an impact on the response to antigen, determining a decrease in the TCR activation as well as the efficacy of CD4+ T cells [167]. Lastly, cell memory generation and the maintenance of CD8+ T cells have also been regulated by autophagy, contributing to the efficacy of antitumour CD8+ T-cell response [168].

## 6. Autophagy and Glioma Stem Cells

Cancer stem cells (CSCs) are a small subpopulation of cancer cells with the abilities of self-renewal and differentiation in different lineages of cancer cells, playing a central role in tumour initiation, progression, and metastatisation [169,170]. Glioma stem cells’ (GSCs) differentiation in differentiated glioblastoma cells causes higher proliferation and tumour recurrence, as well as chemoresistance [171]. However, it has been reported that autophagy plays an important role in the modulation of the CSC population, although its mechanism is still not fully elucidated [172,173]. The high expression of LC3-II, Atg5, and Atg12 has been observed in GSCs with the CD133 marker, exhibiting low phosphorylation of AKT/mTOR, thus correlating with pro-survival autophagic activity [174]. Furthermore, GSCs showed increased or decreased Beclin-1 expression compared to normal glial cells [175]; in detail, higher levels of autophagy regulated GSCs’ maintenance and function, while the inhibition of autophagy significantly depleted the pool of the GSC population [174]. Interestingly, the GSC population increases through the induction of autophagy with TMZ treatment [171,176].

## 7. Resistance to Treatment and Autophagy-Targeting Agents

The controversial role of autophagy in promoting and suppressing tumour growth and its implications in glioma treatment are, to date, a matter of discussion. Some studies have shown that autophagy inhibition increases the cytotoxicity of chemo- and radiotherapy; by contrast, other reports suggested that the activation of autophagy can induce apoptosis and, consequently, the therapeutic efficacy of several treatments [177,178] (Table 1).

The standard treatment for high-grade gliomas is surgical resection of the tumour, ionizing radiation (IR), and the administration of Temozolomide. This treatment (IR + TMZ) showed an increase in the median survival from 12.1 months to 14.6 months, and an increase in the two-year survival rate from 10.4% to 26.5%, with respect to irradiation alone. However, TMZ has a moderate therapeutic outcome due to the occurrence of chemoresistance [255]. However, both TMZ and IR increase autophagic activity through the accumulation of ROS, the triggering of endoplasmic reticulum (ER) stress, and the activation of several signalling pathways, such as the PI3K/AKT/mTOR, ATM/AMPK/ULK1, and JAK2/STAT3 [28,177,256], promoting tumour cells’ survival and chemoresistance. In patients with recurrent glioblastoma and good performance status, Regorafenib can be taken into consideration as a therapeutic option. In particular, the radiosensitivity of the tumour cells is frequently related to the suppression of the autophagic machinery; in fact, nuclear translocation of Beclin1 has been observed in response to IR treatment, while ATG5-driven autophagy has been able to promote the radio-sensitivity of cancer cells [177,178]. On the other hand, if Beclin1 or ATG5 expression have been silenced, a reduction in IR sensitivity in neoplastic elements has been reported [255,256].

It has been demonstrated that Regorafenib steadies PSAT1 to trigger PRKAA-dependent autophagy initiation and suppresses RAB11A-mediated autophagosome–lysosome fusion, as a consequence of the lethal autophagy arrest in GBM cells [257]. HGGs have also been reported to be resistant to other standard therapeutic agents, e.g., Bevacizumab, an inhibitor of VEGF, due to the induction of hypoxia-associated autophagy [258,259]. The induction of apoptosis and the regulation of autophagy could represent an approach to overcoming resistance to glioblastoma therapeutic treatments.

Currently, new autophagy-related agents able to improve the standard treatment are still in the development phase. Among the autophagy inhibitors, Chloroquine (CQ) suppresses autophagy by preventing autophagosome–lysosome fusion by increasing the intralysosomal pH [179]. However, it has been reported that a combination of CQ with standard treatment increases the overall survival of GBM patients and can improve ionizing radiation-induced cell death [179]. In addition, treatment with CQ plus IR increases apoptosis in U87MG cells by decreasing Bcl-2 expression and increasing caspase-3 expression [182]. Moreover, a combination of CQ and TMZ promotes apoptosis in U87MG cells by increasing the mitochondrial ROS [180,181]. According to preclinical trials, GBM patients may benefit from a combination of CQ and chemoradiation, thus making tumour treatment more effective [179]. Tyrosine-kinase inhibitors (TKIs) such as Vandetanib can also induce autophagy in glioma cells by downregulating the PI3K/Akt/mTOR signalling pathway. A combination with CQ and Vandetanib has been reported to inhibit tumour growth in U251 cells, because Chloroquine increases the apoptotic effect of Vandetanib [198].

Erlotinib is an EGFR kinase inhibitor. It has been shown that high concentrations of Erlotinib lead to cell death due to autophagy in the U87-MG glioma cell line. Erlotinib could be combined with other molecules that regulate autophagy and promote apoptosis to deliver Erlotinib in a therapeutic range [201]. The association of Erlotinib and Crizotinib (c-Met inhibitor) determined the induction of apoptosis and a notable decrease in tumour growth in primary human GBM cell models [202]. Furthermore, Erlotinib with NSC23766 (an RAC1 inhibitor) triggered apoptosis and autophagy in human glioma cell lines in vitro [203]. Despite promising results in preclinical studies, the efficacy of Erlotinib has not been shown in clinical trials. In a phase II trial, 110 patients with progressive GBM after prior radiotherapy were randomly assigned to receive Erlotinib, Temozolomide, or Carmustine. The study demonstrated the inefficacy of Erlotinib as a single agent in patients with glioblastoma (the PFS-6 was 11.4% in the Erlotinib group and 24% in the control group) [204]. A phase II trial that evaluated the association of Erlotinib plus Bevacizumab in unmethylated GBM patients after treatment with radiation and Temozolomide did not reach the primary endpoint of increasing survival [205]. Another phase II trial studied the combination of Erlotinib and Sorafenib in patients with recurrent glioblastoma. However, this study also demonstrated the ineffectiveness of Erlotinib to increase survival in patients with glioblastoma [206].

Gefitinib is an EGFR kinase inhibitor. Several studies have shown that Gefitinib induces apoptosis. Instead, Chang et al. proved that lower levels of Gefitinib activated AMPK-dependent autophagy, which inhibited glioma cell growth [207]. Furthermore, the association of Gefitinib and valproic acid triggered autophagy with the activation of the LKB1/AMPK pathway in glioma cells [208]. However, clinical trials have not been as encouraging. Indeed, a phase 1/2 study evaluated the treatment with Gefitinib and radiotherapy in patients with newly diagnosed glioblastoma. The trial failed to prove the therapeutic efficacy of Gefitinib with RT [210]. Another phase II study attempted to demonstrate the efficacy of Gefitinib as a possible therapy in patients newly diagnosed with glioblastoma after radiotherapy, but the trial documented that the therapy with Gefitinib was not related to a significant improvement in the OS or PFS [209].

Imatinib is an inhibitor of several protein tyrosine kinases, such as Abl, c-KIT, and PDGF-R. Imatinib has been shown to induce autophagy in glioma cells [211]. In fact, it increases the phosphorylation of ERK1/2 and suppresses the AKT/mTOR signalling pathway [211]. Preclinical studies demonstrated that Imatinib could be considered as a therapeutic choice in glioblastoma. However, even though clinical trials were initiated, they showed that Imatinib had no significant activity in patients with newly diagnosed or recurrent glioblastoma [212,213].

Sorafenib is a multitarget TKI that inhibits VEGFR2/3, PDGFR, FLT3, c-KIT, and the RAF/MEK/ERK pathway. Sorafenib has been shown to induce autophagy in glioblastoma cells. However, the association of Sorafenib and Temozolomide suppressed autophagy and induced apoptosis [214,215]. In fact, it has been shown that the association between Sorafenib and molecules that inhibit autophagy causes an increase in the antineoplastic activity of Sorafenib in glioma cells [214,215]. Furthermore, there is evidence that the combination of Sorafenib with Lapatinib determines the induction of autophagy and the cell death of glioblastoma [216]. Several clinical trials have been conducted to evaluate the association of Sorafenib with TMZ. In particular, a phase II study demonstrated the efficacy of this association; in fact, the primary end point of the trial was achieved with a PFS of 26% [217].

Lonafarnib is an oral small-molecule inhibitor of farnesyltransferase. In cancer cells, Lonafarnib promotes autophagy by causing an alteration in the Ras/PI3K/AKT/Rheb/mTOR pathway [218]. In preclinical studies, Lonafarnib was effective in inhibiting cell growth in glioblastoma. Therefore, a phase I/Ib study was conducted to assess the association between Lonafarnib and Temozolomide in patients with recurrent glioblastoma, demonstrating the efficacy of this therapy in this setting of patients [219].

Vemurafenib is an ERK inhibitor in BRAFV600-positive tumour cells. Brain tumour cells with the BRAFV600E mutation have increased autophagy activity. The combination of Vemurafenib with Chloroquine can improve autophagy inhibition and determine cell death [220].

3-Methyladenine (3-MA) acts as an autophagy suppressor in a nutrient-poor environment through the inhibition of PI3KC3 [195]. In U251 human glioma cells, 3-MA can augment cisplatin-induced apoptosis by increasing ER stress [196]. In combination with melatonin, 3-MA can diminish Bcl-2 expression and increase Bax expression by suppressing autophagy and favouring apoptosis in U87 glioma cells [197].

Quinacrine (QC) can increase TMZ toxicity by inducing apoptosis [185]. The efficacy of QC can be improved by hypoxia, causing an increase in ATG LC3-II expression and apoptosis [186,187]. In addition, a combination of QC and Cediranib can inhibit AKT phosphorylation in GBM cells [187].

ABT-737 has been reported to promote apoptosis in U87 and U251 glioma cells, increasing Bax expression but neutralizing Bcl-2 and autophagic flux [222]. Consequently, it has been suggested that ABT-737 may be employed as a single-agent treatment to sensitize glioblastoma cells to TMZ [260].

The inhibition of autophagy enhances the apoptosis induced by the proteasome inhibitor bortezomib (BTZ) in human glioblastoma U87 and U251 cells [223,224]; moreover, clinical trials have shown that a combination of BTZ and standard treatment improved the overall survival to 19.1 months [226] through the inhibition of the NK-κB pathway, making GBM cells more sensitive to TMZ [225].

Sirolimus (rapamycin), Temsirolimus, and Everolimus are autophagy inducers that act by inhibiting the PI3K/AKT/mTOR pathway [177]. Sirolimus, as a single agent or in combination with Erlotinib, reduces cell proliferation by blocking the mTOR pathway in U87 glioma cells and reduces tumour size in mouse models [227,228]. Moreover, Sirolimus induces autophagy in GSCs and promotes the differentiation of these cells, both in vitro and in vivo [229]. It has been reported that Sirolimus favours radiosensitivity [261] and, in combination with TMZ and CQ, induces apoptosis in U87 glioma cells through the release of cathepsin B [230]. Temsirolimus blocks mTOR activation by inhibiting HIF-1α and the expression of VEGF [231,232]; in addition, preclinical trial findings demonstrated that Temsirolimus improved the efficacy of IR or TMZ in recurrent glioblastomas, as documented by imaging improvements [262]. Everolimus, a Rapamycin analogue, can suppress angiogenesis and promote autophagy by inhibiting mTOR [263]. Specifically, the Atg5 expression in U87 glioma cells is activated by Everolimus, reducing glioma cell proliferation and increasing the median survival, as documented in preclinical and clinical trials [236]. Finally, a combination treatment of Everolimus and TMZ can increase TMZ efficacy in glioma cells [263,264].

Momelotinib (MTB) is an inhibitor of JAK1/2 that inhibits tumour growth in U251 glioma cells through the upregulation of autophagy-related proteins’ expression, such as LC3, Beclin-1, and p62 [240]. However, a combination of MTB and TMZ reduces the phosphorylation of AKT and STAT3 [240].

Metformin inhibits cell proliferation and promotes autophagy and apoptosis through the inhibition of mTORC1 [241]. A combination of metformin and IR or TMZ can modulate apoptosis by increasing the Bax expression and can sensitize glioma cells to standard treatment [242,243]. A co-treatment of metformin with arsenic trioxidecan helps GSCs differentiate into nontumorigenic cells [235]; in detail, metformin works by activating the AMPK-FOXO3 axis, whereas arsenic trioxide inhibits the phosphorylation of STAT3 caused by IL-6 [244].

A combination of Simvastatin and TMZ has been reported to promote apoptosis in U251 cells though the inhibition of the autophagosome–lysosome fusion [248]; Lovastatin has also been shown to be able to reduce autophagic activity through the inhibition of the ATK/mTOR pathway [249].

Perifosine is an alkyl phospholipid that inhibits cell proliferation in GBM patients treated with TMZ [265], acting on the inhibition of AKT/mTOR with the interference in the recruitment of AKT to the plasma membrane [266,267,268]. Moreover, it improves the efficacy of Bevacizumab, resulting in antiproliferative activity and a longer survival rate [269]. It has been reported that the toxicity of Perifosine in GBM cells is improved by the combination with short-chain cell-permeable ceramide (C6).

Suberoylanilide hydroxamic acid (SAHA) induces autophagic activity and inhibits tumour growth by attracting LC3-II to the autophagosome membrane and increasing the formation of autophagosome vesicles [188,189]; it also increases the BeCN1 levels and lowers the SQSTM1 levels [190]. SAHA also prevents cell invasion in glioma and reorganizes intratumoral TME [188,189]. Furthermore, SAHA can cause apoptosis in GSCs and activation of the caspase-8- and -9-mediated pathways [191]. Interestingly, a lower dose of SAHA can inhibit GSCs by activating cell cycle arrest and causing premature senescence through p53 and p38 induction [191].

Imipramine is a tricyclic antidepressant able to promote autophagy in astrocytes and neurons [250]. In U87 glioma cells, Imipramine inhibits the PI3K/AKT/mTOR signalling pathways and enhances the conversion of LC3-I in LC3-II for autophagosome formation [251]. Additionally, it can inhibit the ERK/NK-κB pathway, blocking the glioblastoma progression [251].

Micro-RNAs (miRNAs) are endogenously expressed through 18-25 noncoding RNAs that regulate autophagy, cell proliferation, angiogenesis, metastatisation, and drug resistance through the silencing of gene expression [270]. In U251 glioma cells, miR-30a increases the chemosensitivity to TMZ by targeting Beclin-1 and preventing autophagy [252]. Autophagy can be induced with miR-128 by inhibiting mTOR and promoting apoptosis through the activation of caspase 3-9 [253]. The efficacy of TMZ can be increased with miR-519a in chemoresistant U87 glioma cells, and autophagy can be induced by modulating the STAT3/Bcl2 pathway [254]. Autophagic activity can regulate miR-93 in GSCs by inhibiting multiple autophagy regulators, including ATG4B, ATG5, BECN1/Beclin-1, and SQSTM1/p62, thus enhancing IR and TMZ activity against GSCs [271].

Together with the well-known capability of autophagy to degrade cellular and sub-cellular components, its interactions with immune processes, including the production of inflammatory cytokines, as well as antigen processing, may increase the host immune defence in order to eliminate pathogens. In this scenario, viruses may also interact with the autophagy cascade with either a favourable or detrimental effect on them [272]. The regulation of autophagy by oncolytic virus infection and the action of the viruses on the cellular autophagy are still debated [272]. Nevertheless, following the increased curiosity for oncolytic viruses (OVs) to clinical development, drug delivery based on an approach characterized by enhanced OV delivery, including the use of nanoparticles as well as complex viral–particle ligands, is under consideration [272].

## 8. Conclusions

The autophagy machinery was analysed in terms of its physiological and pathological characteristics, allowing for a better definition of the molecular mechanisms governing this process. The identification of ATG genes and proteins enables to one understand the complexity of the autophagy pathways and its impact on human health. We focused on the role of autophagy in gliomas, analysing its dual action both as a tumour suppressor and as a tumour promoter. Hypotheses concerning the different levels of ATG proteins in the transition from low- to high-grade gliomas were proposed with reference to their prognostic value. Moreover, the relationships between autophagy and TIME in gliomas were also discussed, taking into consideration the different cellular contribution in the autophagic mechanism. After a brief discussion about autophagy and glioma stem cells, new developments of autophagy-targeting agents were described and their combinations, including the performed clinical trials. It should be mentioned that, in addition to chloroquine, drugs with higher selectivity for autophagy are developing and hopefully applied to clinical practice. Finally, novel methods and high-throughput technologies should be used to understand autophagy in gliomas, mainly by the identification of targets utilizing clustered regularly interspaced short palindromic repeats (CRISPRs)—CRISPR-associated protein (Cas9) or CRISPR-Cas9 genome editing and/or application of miRNAs (Table 2). Their utilization for a synergistic combination with TMZ in the context of inhibition of autophagy in human gliomas should be desirable. Consequently, this modern perspective could help in the selection of patients with gliomas that are most likely to respond to autophagy inhibition therapy but also to identify patients resistant to treatment.

## Figures and Tables

**Figure 1 cancers-15-02622-f001:**
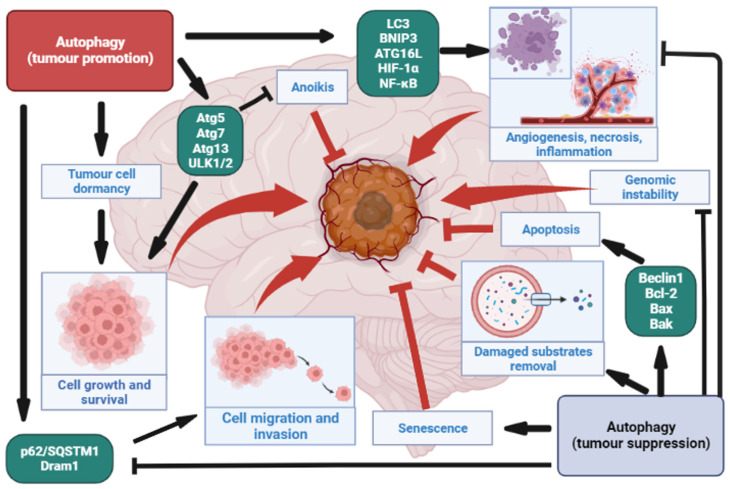
Different biological effects of autophagy in gliomas are schematically identified.

**Figure 2 cancers-15-02622-f002:**
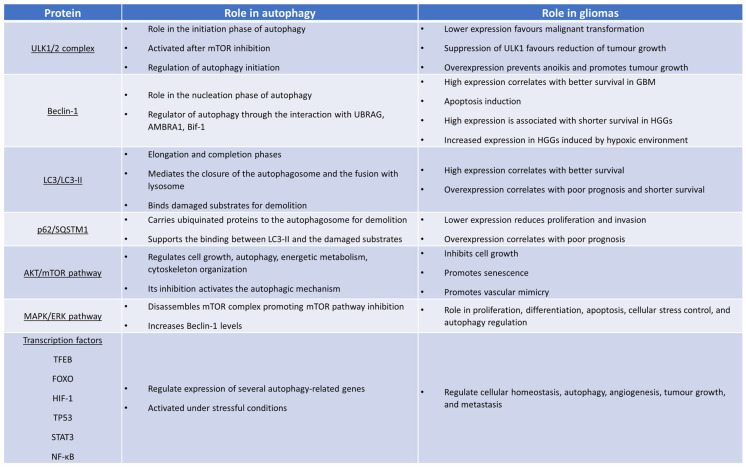
The role of the most relevant autophagy-involved agents in gliomas.

**Table 1 cancers-15-02622-t001:** Autophagy-related agents with their direct/indirect effects and corresponding clinical trials.

Direct Effects on Autophagy Pathway		
**Agent**	**Mechanism of action**	**Clinical trials targeting autophagy in glioma**
Choloquine (CQ)	Suppresses autophagy by preventing autophagosome-lysosome fusion [179] CQ + TMZ promotes apoptosis in U87MG cells by increasing mitochondrial ROS [180,181] CQ + IR increases apoptosis in U87MG by decreasing Bcl-2 expression and increasing caspase-3 expression [182]	Median OS 24 months compared to 11 months for control group [183] CQ + RT + TMZ: Median OS 24 months [184] CQ + RT + TMC: Median OS was 11.5 months for EGFRvIII-patients and 20 months for EGFRvIII + patients [179]
Quinacrine (QC)	Increases TMZ toxicity by inducing apoptosis [185] Increases LC3-II expression and apoptosis in hypoxic environment [186,187] QC + Cediranib inhibits AKT phosphorylation in GBM cells [187]	No clinical trials available
Suberoylanilide Hydroxamic Acid (SAHA)	Induces autophagy by increasing autophagosome formation vesicles [188,189] Increases BeCN1 levels and decreases SQSTM1 levels [190] Induces apoptosis of GSCs by activating caspase8 and -9 madiated pathways [191]	Median OS 5.7 months [192] SAHA + Bevacizumab + TMZ: no significant improvement of 6-months survival time [193] SAHA + RT : no improved outcome [194]
3-Methyladenine (3-MA)	Suppresses autophagy by inhibiting PI3KC3 [195] Promotes cisplatin-induced apoptosis by increasing ER stress in U251 cells [196] 3-MA + melatonin suppresses autophagy and increases apoptosis by increasing Bax expression in U87MG cells [197]	No clinical trials available
**Agent**	**Indirect effects on autophagy**	**Clinical trials targeting autophagy in glioma**
Vandetanib Tyrosine-kinase inhibitors (TKI)	Induces autophagy in glioma cells by downregulating the PI3K/Akt/mTOR signalling pathway [198] Vandetanib + CQ inhibits tumour growth in U251 cells [198]	Median OS 6.3 months in recurrent GBMs [199] Vandetanib + RT/TMZ: no significant improved survival compared to control group [200]
Erlotinib EGFR kinase inhibitor	Promotes autophagic cell death in U87MG cells [201] Erlotinib + Crizotinib induces apoptosis and decreases tumour growth in human glioma cells [202] Erlotinib + NSC2376 induces apoptosis and autophagy in human glioma cells [203]	No significant improvement in OS [204] Erlotinib + bevacizumab: median OS 13.2 months; no increased survival [205] Erlotinib + sorafenib: median OS 5.7 months; no increased survival [206]
Gefitinib EGFR kinase inhibitor	Inhibits glioma cell growth by activating AMPK-dependent autophagy [207] Gefitinib + valproic acid induces autophagy by activating the LKB1/AMPK pathway in glioma cells [208]	OS at 1 year similar to control group [209] Gefitinib + RT: median survival 11.5 months; no significant improvement compared to control group [210]
Imatinib Tyrosine-kinase inhibitors (TKI)	Induces autophagy in glioma cells by increasing the phosphorylation of ERK1/2 and suppressing the AKT/mTOR signalling pathway [211]	Median OS 5-6.5 months; no significant improvement of OS [212,213]
Sorafenib Tyrosine-kinase inhibitors (TKI)	Induces autophagy by inhibiting VEGFR2/3, PDGFR, FLT3, c-KIT, and the RAF/MEK/ERK pathway [214,215] Sorafenib + TMZ suppresses autophagy and induces apoptosis [214,215] Sorafenib + Lapatinib induces autophagy and cell death of GBM [216]	Sorafenib + TMZ: improvement of PFS (26%) [217]
Lonarfanib	Promotes autophagy by causing an alteration in the RAS/PI3K/AKT/Rheb/mTOR pathway [218]	Lonarfanib + TMZ: PFS at 6 months 38%, median PFS 3.9 months, median disease-specific survival 13.7 months [219]
Vemurafenib	Increases autophagy activity in BRAFV600E-positive tumour cells [220] Vemurafenib + CQ improves autophagy inhibition and promotes cell death [220]	OS 11.9 months, PFS 5.3 months [221]
ABT-737	Promotes apoptosis in U87 and U251 glioma cells by increasing Bax expression and decreasing Bcl-2 and activates autophagy [222]	No clinical trials available
Bortezomib	Induces apoptosis by inhibiting autophagy in U87 and U251 glioma cells [223,224] Improves TMZ sensitivity by inhibiting NK-κB pathway [225]	Bortezomib + TMZ + RT: OS 19.1 months [226]
Sirolimus (rapamycin)	Reduces cell proliferation in U87MG by blocking mTOR pathway [227,228] Induces autophagy in GSCs [229] Sirolimus + TMZ and CQ induces apoptosis in U87MG by releasing cathepsin B [230]	
Temsirolimus	Blocks mTOR activation by inhibiting HIF-1α and VEGF expression [231,232]	Temsirolimus + erlotinib: no therapeutic effect due to high toxicity [233] Temsirolimus + Bevacizumab: radiological stable disease, PFS 8 weeks, OS 15 weeks [234] Temsirolimus + sorafenib: no improvement of OS [235]
Everolimus	Suppresses angiogenesis and promotes autophagy by inhibiting mTOR [236] Reduces U87MG proliferation by increasing Atg5 expression [236]	Everolimus + RT/TMZ: no improvement of OS [237] Everolimus + Bevacizumab + RT/TMZ: OS 13.9 months, PFS 11.3 months [238] Everolimus + gefitinib: no therapeutic effect [239]
Momelotinib (MTB)	Inhibits tumour growth in U251 cells by upregulating LC3, Beclin-1 and p62 expression [240] MTB + TMZ reduces the phosphorylation of AKT and STAT3 [240]	No clinical trials available
Metformin	Inhibits cell proliferation and promotes autophagy by inhibiting mTORC1 [241] Metformin + IR/TMZ modulates apoptosis by increasing Bax expression [242,243] Metformin + arsenic trioxidecan drives GSCs into nontumorigenic differentiation by activating AMPK-FOXO3 and inhibiting STAT3 [244]	Median OS 19.9 months [245] Metformin + RT: Median PFS 10 months for newly diagnosed GMs and 4 months for recurrent GBMs [246] No improved OS in newly diagnosed GBMs [247]
Simvastatin	Simvastatin + TMZ promotes apoptosis in U251 cells by inhibiting the au-tophagosome-lysosome fusion [248]	No clinical trials available
Lovastatin	Reduces autophagy by inhibiting the AKT/mTOR pathway [249]	
Imipramine	In U87 cells inhibits PI3K/AKT/mTOR pathway and enhances the conversion of LC3-I in LC3-II [250] Inhibits ERK/ NK-κB pathway blocking GBM progression [251]	Ongoing study (NCT04863950)
Micro-RNAs (miRNAs)	miR-30a increases TMZ sensitivity by targeting Beclin-1 and preventing autophagy in U251 cells [252] miR-128 induces autophagy by inhibiting mTOR and promoting apoptosis by activating caspase 3-9 [253] miR-519a induces autophagy by modulating STAT3/Bcle2 pathway [254]	No clinical trials available

**Table 2 cancers-15-02622-t002:** Targets to understand autophagic ATGs involvement in GBM either by CRISPR-Cas9 or miRNA.

CRISPR-Cas9 Genome Editing Application in GBM	miRNA	Target (s)
ATM	miR-93	Beclin 1, ATG5, ATG4B, SQSTM1/p62
ATG5	miR-30a	Beclin 1
ATG7	miR-224-3p	ATG5
TSC2	miR-17	ATG7
	miR-224-3p	ATG5,
	miR-7-1-3p	mTOR, SQSTM1, p62
	miR-138	LC3-II, BIM
	miR-30e	Beclin-1
	miR-590-3p	LC3-II, Beclin-1,
	miR-155-3p	LC3B-II, SQSTM1

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
