# Peer review of "Autophagic-Related Proteins in Brain Gliomas: Role, Mechanisms, and Targeting Agents"

_cancers, 2023, doi:10.3390/cancers15092622_

Round 1
Reviewer 1 Report
In this manuscript by Pizzimenti et al., the authors have discussed the role of autophagy in regulating the mechanisms and therapeutic targets in gliomagenesis. In this review the authors have provided a detailed overview of the process and mechanisms of autophagy in general followed by the role of autophagy in gliomas, wherein they have discussed the dual role of autophagy in glioma by acting as an oncogenic regulator as well as a tumor suppressor. Finally they have discussed the role of autophagy in regulating tumor resistance in glioma and the drugs in clinical trials that targets the autophagic proteins with a role in gliomagenesis.
In the sections where the authors discuss the dual role of autophagy in gliomagenesis, a perspective on the dual role is missing. What factors drive the two way regulation of autophagy in cancer and in glioma? Since autophagy is a cellular defense mechanism, how is the balance disturbed and pivoted in favor of brain tumor progression? Moreover, in the table provided in Figure 3, the authors provide both poor and better survival results for high expression of proteins such as LC3 and similar data for the other proteins. The authors should provide a perspective on this kind of data as it might not be useful for the reader without a point of view. Finally, in the resistance section, the authors should mention all the drugs described in inhibiting autophagic processes in the form of a table with relevant information such as mechanisms, targets etc. Figure 1 lacks novelty and should rather provide regulation of autophagic processes in context of glioma providing the most recent methods and high throughput technologies used to study autophagy in glioma. The current enthusiasm for this manuscript is moderate.
Author Response
First of all, we wish to thank to reviewers for their useful and appreciable suggestions. In detail:
Rev 1 - we have addressed the requested major issues, changing the manuscript and underlining sections with yellow ink:
- “In the sections where the authors discuss the dual role of autophagy in gliomagenesis, a perspective on the dual role is missing. What factors drive the two way regulation of autophagy in cancer and in glioma?
To better explain the dual role we have added the following sentences: “Autophagy has been known to have a dual role in promoting or suppressing tumour initiation and growth in different typed of cancers, gliomas included [1,17-18]. Specifically, autophagy may act as a tumour promoter by recycling substrates for sustaining tumour metabolism and neoplastic survival under adverse circumstances, such as hypoxic stress or nutrient deprivation [19,20]. Alternatively, autophagy show a role of tumour suppressor and inhibitory function by removing damaged substrates and organelles, protecting also cells from ROS, inflammation, necrosis and other causes of genomic instability [21,22]. Several studies explored the dual role of autophagy in gliomas with controversial results as reported in the following subheadings”.
- Since autophagy is a cellular defence mechanism, how is the balance disturbed and pivoted in favour of brain tumour progression?”
To better explain the disturbed balance and tumor progression we have added the following sentences: “It has been suggested that one of the mechanisms able to produce a disturb and pivotal role in favor of brain tumor progression may be represented by hypoxia, determining the activation of the hypoxia-inducible factor 1-alpha (HIF-1α), which induces autophagy through the transcription regulation of autophagic genes [120-123]. In addition, HIF-1α also stimulates angiogenesis to have more oxygen and nutrients available for the survival of neoplastic elements by VEGF upregulation [124,125]. However, the rate of hypoxia and the expression of angiogenic factors may be directly related to tumor grade as well as worst prognosis in brain human gliomas”.
- “The authors should provide a perspective on this kind of data as it might not be useful for the reader without a point of view.”
We have added the following sentences: “At the end of this section, our point of view suggests that explanations for the controversies whether the autophagy pathway promotes survival or death are still debated. In fact, the balance between pro-survival or pro-death autophagic factors may be strongly related to their relationships since low to moderate levels of autophagy activation become cytoprotective, while high autophagic levels develop cytotoxicity”.
- “In the resistance section the authors should mention all the drugs described in inhibiting autophagic processes in the form of a table with relevant information such as mechanism, targets etc.”
As requested we have added a new table with relevant information about targets and mechanisms of mentioned drugs”.
- “Figure 1 lacks novelty and should rather provide regulation of autophagic processes in context of glioma providing the most recent methods and high throughput technologies used to study autophagy in glioma.”
We have restyled Fig. 1 and we have added some comments about methodologies and technologies used to study autophagy in gliomas inside the conclusion section.
We hope that in the present form our manuscript may be suitable for the publication in Your special issue of prestigious Cancers Journal.
Sincerely Yours,
Reviewer 2 Report
This manuscript has so many language problems (see sample from page one only!) that it is a pain to read. As the next step, either a native speaker of the English language with expertise in the field should correct the paper or a commercial language editing service. It does not make sense in my opinion to continue the review process without that having been done in great detail.

Author Response
First of all, we wish to thank to reviewers for their useful and appreciable suggestions.
As requested we have provided an extensive certificated English revision performed by MDPI Editing Service.
We hope that in the present form our manuscript may be suitable for the publication in Your special issue of prestigious Cancers Journal.
Sincerely Yours,
Reviewer 3 Report
Dear Authors,
I have read your manuscript entitled "Autophagic Related Proteins in Brain Gliomas: Role, 2 Mechanisms and Future Therapeutic Insights" with interest, and hereby I am sharing with you my suggestions for improvement.
I would like to start with a reflection about the (very extensive, congratulations) bibliography: would you consider including a comment about how did you select the bibliographic items to be included in this review, and whether this review was ever meant to be exhaustive and systematic (thus mentioning the search terms, the inclusions and exclusion criteria, and the dates at which your search is last updated), or narrative? This could be added at the end of the Introduction (and mentioned in the abstract), and would make it clearer to the reader what to expect.
I have enjoyed the presence of schemes and tables to resume and highlight the fundamental contributions of the major sections of the review, and this made me feel the emptiness left by the absence of an analogous representation for the section #7 (Resistance to treatment and autophagy-targeting agents), despite the declared take home message of the review according to your own words in the conclusions (see lines 520-524) would be knowledge aimed at helping "in the selection of patients with gliomas that are most likely to respond to autophagy inhibition therapy, but also to identify patients resistant to treatment".
To the same end, that of clinical translation, the review describes as a handbook would the available literature, but steers away from explicitly identifying those instances about which literature has good agreement, or it is still merely exploratory, or it is in competing disagreement between models/interpretations. A tissue/metabolic map, or a table, sharing with the readers information about uncertainty of the presented information, and pinpointing open questions, would multiply the potential impact of the review work, with little extra effort by a team that has already gone through and organized all that literature.
Best regards
Author Response
Dear Reviewer,
First of all, we wish to thank to reviewer for their useful and appreciable suggestions. In detail:
We have added at the end of the introduction and in the abstract the astrocytic lineage to make clearer the inclusion and exclusion criteria of bibliographic selection.
We have changed the tables and figures focusing the attention of the reader to different biological effects of autophagy in gliomas, the role of the most relevant ATGs in gliomas and the drug agents involved in autophagy with their targets and mechanisms.
We hope that in the present form our manuscript may be suitable for the publication in Your special issue of prestigious Cancers Journal.
Sincerely Yours,
Reviewer 4 Report
This review article by Pizzimenti et al. focuses on the role and therapeutic implications of autophagy-related proteins in gliomas. While this is an interesting topic, the selection of main questions that are discussed appears somewhat arbitrary and limited. Based on this assessment, the manuscript would benefit from additional important information that should be incorporated to further improve its impact.
A state of the art overview of ongoing and finished clinical trials targeting autophagy in brain tumors (chloroquine, hydroxychloroquine, repurposed drugs etc.) would be useful. This could be shown in a separate table with an accompanying chapter in the text.
A general overview of autophagy pathway is ok (Fig. 1), but can be found in a plethora of other articles. A well-prepared figure illustrating and combining the different biological effects of autophagy in gliomas and in their microenvironment would be a very nice addition. Fig.2 provides only text bullet points.
Fig. 3 is also rather minimalistic. The authors could try to integrate data on the activation status of other proteins/pathways that are known as essential regulators of autophagy, e.g. mTOR, p53.
What is the effect of radiation therapy on autophagy?
What is the role of chaperone-mediated autophagy in brain tumors?
Glioblastoma appears to be particularly vulnerable to autophagic cell death. The dual role of autophagy in promoting cell survival vs (autophagic) cell death should be explained and attempts to exploit pro-death autophagy for glioma treatment integrated in the article.
Please also integrate information on autophagy and oncolytic viruses and nanoparticle-based therapy.
What is known on selective types of autophagy in gliomas (mitophagy, ER-phagy etc.)?
In general, therapeutic strategies focusing mainly on the core autophagy pathway (e.g. chloroquine) should be discriminated from strategies also aimed at targeting other pathways/biological effects (e.g. mTOR inhibition) throughout the article.
Author Response
Dear Reviewer,
First of all, we wish to thank to reviewer for their useful and appreciable suggestions. In detail:
We have added a specific table concerning the drugs utilized in clinical trials, underlying target and mechanisms with corresponding references.
We have restyled Fig. 1 and omitted the Fig. 2 presenting only text bullet points. Moreover, Fig. 3, already considered rather minimalistic, has been changed in Fig.2, in which the most relevant ATGs as well as essential regulators of autophagy (mTOR, p53) have been added.
We have added a paragraph concerning the effect of radiation therapy on autophagy (line 381-386)
We have added a paragraph concerning the role of chaperone-mediated autophagy in brain tumors (line 204-207)
After to analyze the dual role of autophagy in gliomas, we have integrated information regarding the action oncolytic viruses and nanoparticle-based therapy (line 552-559).
Concerning selective types of autophagy in gliomas (mitophagy, ER-phagy etc.), we have added a paragraph (line 208-216).
We have realized a specific table regarding the drug treatments with their mechanism of action.
We hope that in the present form our manuscript may be suitable for the publication in Your special issue of prestigious Cancers Journal.
Sincerely Yours,
Round 2
Reviewer 2 Report
There are still problem areas which I attach to make it easier for the authors. Please eliminate the mistakes the yellow sections in the attach file.

Author Response
Revised version Manuscript n. cancers-2298054
MESSINA, 27 th April 2023
Estimed Guest Editor: Prof. Dr. Gabriella D’Orazi,
Dear Reviewers,
we send You the newly revised version of our manuscript after having done the changes requested by reviewer 2.
In detail:
There are still problem areas which I attach to make it easier for the authors. Please eliminate the mistakes the yellow sections in the attach file.
As requested by the reviewer the sentences underlined by red ink have been modified as follows:
Sentence line 203 has been erased
Sentence line 204 has been changed
Sentence line 208 has been changed
Reviewer 4 Report
Pizzimenti et al. provide a revised version of their manuscript focusing on the role and therapeutic implications of autophagy-related proteins in gliomas. The authors incorporated additional information to improve the timeliness of the article, but unfortunately, the results are not entirely successful and the changes often somewhat miss the point regarding the questions raised.
Instead of an overview of ongoing and finished clinical trials targeting autophagy in brain tumors as requested, the authors provide a new table on drugs utilized in clinical trials in general, with the underlying target mechanisms derived from preclinical studies. The current clinical status of exploiting autophagy for the treatment of brain tumors is an important point that needs to be addressed. Accordingly, searching the Clinicaltrials.gov site, e.g. with “glioblastoma” and “autophagy” will give several hits. This should be shown in a separate table or be incorporated as a separate section (clinical trials targeting autophagy in glioma) into the current table. As stated before, therapeutic strategies focusing mainly on the core autophagy pathway (e.g. chloroquine) should be discriminated from strategies also aimed at targeting other pathways/biological effects (e.g. mTOR inhibition) throughout the article. This information should also be incorporated into the table, e.g. by introducing separate sections. This organization (pre-clinical vs clinical, direct autophagy targeting vs other pathways) would also help to improve Chapter 7 which is a rather unstructured listing of different compounds at the moment.
As already stated, a well-prepared figure illustrating and combining the different biological effects of autophagy in gliomas and in their microenvironment would be a very nice addition. Unfortunately, the provided quality of new Fig. 1 is sub-par and needs to be improved significantly to make it more comprehensible and to better illustrate matters, e.g. using a software such as BioRender or Smart Servier Medical Art. Since the article lacks high quality and catchy figures in its present state, it perhaps would also be a feasible idea to provide two separate figures, one focusing on the role of autophagy (factors) in glioma biology (tumor initiation/growth, microenvironment etc.) and one on targeting autophagy in glioma, illustrating the mode of interference on the cellular level/in the TME. In any case, high quality figures would definitely improve the overall quality of the article.
Fig. 2 providing information on the general role of autophagy factors in gliomas has been extended. Only the minority of proteins are ATGs, so please correct the legend.
The authors somewhat missed the point regarding the dual role of autophagy in promoting cell survival vs (autophagic) cell death, a topic that needs to be better explained. Glioblastoma appears to be particularly vulnerable to autophagic cell death, as e.g. demonstrated by the work of Hanahan (PMID: 26412325), Velasco (PMID: 19425170) and other groups. Attempts to exploit pro-death autophagy for glioma treatment therefore should be better integrated in the article. The new sentence “the balance between pro-survival or pro-death autophagic factors….” misses the point because cell-lethal excessive autophagy rather reflects enforced, drug-induced overactivation of autophagy, not an imbalance of autophagy factors in the tumors.
The authors have now incorporated a new part on selective autophagy. The sentence “…reverted cannabidiol-induced glioma cell death, suggesting the favourable role of mitophagy on anti-tumor” is incomplete. Also in this section, the role of selective types of autophagy for tumor biology and therapy need to be discriminated, including the potential role of selective autophagy in cell survival vs death (e.g. PMID: 33111629; PMID: 29938581).
Regarding the outlook, it should be mentioned that in addition to chloroquine, drugs with higher selectivity for autophagy are currently developed that will hopefully be transferred to the clinic at some point.
Author Response
Revised version Manuscript n. cancers-2298054
MESSINA, 27 th April 2023
Estimed Guest Editor: Prof. Dr. Gabriella D’Orazi,
Dear Reviewers,
we send You the newly revised version of our manuscript after having done the changes requested by reviewer 4 (in red ink).
In detail:
Pizzimenti et al. provide a revised version of their manuscript focusing on the role and therapeutic implications of autophagy-related proteins in gliomas. The authors incorporated additional information to improve the timeliness of the article, but unfortunately, the results are not entirely successful and the changes often somewhat miss the point regarding the questions raised. Instead of an overview of ongoing and finished clinical trials targeting autophagy in brain tumors as requested, the authors provide a new table on drugs utilized in clinical trials in general, with the underlying target mechanisms derived from preclinical studies. The current clinical status of exploiting autophagy for the treatment of brain tumors is an important point that needs to be addressed. Accordingly, searching the Clinicaltrials.gov site, e.g. with “glioblastoma” and “autophagy” will give several hits. This should be shown in a separate table or be incorporated as a separate section (clinical trials targeting autophagy in glioma) into the current table. As stated before, therapeutic strategies focusing mainly on the core autophagy pathway (e.g. chloroquine) should be discriminated from strategies also aimed at targeting other pathways/biological effects (e.g. mTOR inhibition) throughout the article. This information should also be incorporated into the table, e.g. by introducing separate sections. This organization (pre-clinical vs clinical, direct autophagy targeting vs other pathways) would also help to improve Chapter 7 which is a rather unstructured listing of different compounds at the moment.
Response: According to reviewer’s suggestions we have incorporated in table 1 as a separate section, the clinical trials targeting autophagy in glioma, also focusing the direct/indirect effects for each therapeutical agent.
As already stated, a well-prepared figure illustrating and combining the different biological effects of autophagy in gliomas and in their microenvironment would be a very nice addition. Unfortunately, the provided quality of new Fig. 1 is sub-par and needs to be improved significantly to make it more comprehensible and to better illustrate matters, e.g. using a software such as BioRender or Smart Servier Medical Art. Since the article lacks high quality and catchy figures in its present state, it perhaps would also be a feasible idea to provide two separate figures, one focusing on the role of autophagy (factors) in glioma biology (tumor initiation/growth, microenvironment etc.) and one on targeting autophagy in glioma, illustrating the mode of interference on the cellular level/in the TME. In any case, high quality figures would definitely improve the overall quality of the article.
Response: According to reviewer’s suggestions we have improved the Figure 1 with a new high quality and more comprehensible picture using the software BioRender.
Fig. 2 providing information on the general role of autophagy factors in gliomas has been extended. Only the minority of proteins are ATGs, so please correct the legend.
Response: According to reviewer’s suggestions we have corrected the legend of figure 2.
The authors somewhat missed the point regarding the dual role of autophagy in promoting cell survival vs (autophagic) cell death, a topic that needs to be better explained. Glioblastoma appears to be particularly vulnerable to autophagic cell death, as e.g. demonstrated by the work of Hanahan (PMID: 26412325), Velasco (PMID: 19425170) and other groups. Attempts to exploit pro-death autophagy for glioma treatment therefore should be better integrated in the article. The new sentence “the balance between pro-survival or pro-death autophagic factors….” misses the point because cell-lethal excessive autophagy rather reflects enforced, drug-induced overactivation of autophagy, not an imbalance of autophagy factors in the tumors.
Response: According to reviewer’s suggestions we have added the following paragraph: . It has been previously argued that cell lethal excessive autophagy reflects enforced, drug-induced, overactivation of autophagy rather than an imbalance of autophagic factors in gliomas (Salazar M et al 2009 doi.org/10.1172/JCI37948). In particular, it has been shown that cannabinoids as well as antidepressant, may induce cell death of cancer cells by autophagic activation, although nontransformed astrocytes appear resistant to cannabinoid killing action (Salazar M et al 2009 doi.org/10.1172/JCI37948; Shchors K et al, 2015 doi.org/10.1016/j.ccel.2015.08.012).
The authors have now incorporated a new part on selective autophagy. The sentence “…reverted cannabidiol-induced glioma cell death, suggesting the favourable role of mitophagy on anti-tumor” is incomplete. Also in this section, the role of selective types of autophagy for tumor biology and therapy need to be discriminated, including the potential role of selective autophagy in cell survival vs death (e.g. PMID: 33111629; PMID: 29938581).
Response: According to reviewer’s suggestions we have added the following paragraph: it has been suggested that early mitochondrial dysfunction and HMOX1 activation may synergize to trigger lethal mitophagy, contributing to the cell death effects of natural compound AT 101 in glioma cells (Meyer N et al. 2017 doi.org/10.1080/15548627.2018.1476812) .In addition, ER-phagy is essential for proliferation and clonogenicity of mutant IDH1 gliomas by downregulation of phospholipid biosynthesis [21,27]. Moreover, autophagic cell death may be triggered by loperamide (LOP) in glioblastoma cells (Zielke S et al, 2020 doi.org/10.1080/15548627.2020.1827780). In detail, LOP may also induce engulfment of large ER fragments within autophagosomes and lysosomes as documented by morphological microscopic investigations (Zielke S et al, 2020 doi.org/10.1080/15548627.2020.1827780). LOP-induced reticulophagy and cell death are predominantly mediated through the reticulophagy receptor RETREG1/FAM134B and, to a lesser extent, TEX264, confirming that ER-phagy receptors can promote autophagic cell death (Zielke S et al, 2020 doi.org/10.1080/15548627.2020.1827780).
Regarding the outlook, it should be mentioned that in addition to chloroquine, drugs with higher selectivity for autophagy are currently developed that will hopefully be transferred to the clinic at some point.
Response: According to reviewer’s suggestions we have added the following paragraph: After a brief discussion regarding autophagy and glioma stem cells, new developments of therapeutic agents targeting autophagy were described for each drug and the combinations of them, also including the performed clinical trials. It should be mentioned that in addition to chloroquine, drugs with higher selectivity for autophagy are currently developed that will hopefully be transferred to the clinical practice.
Round 3
Reviewer 2 Report
The manuscript has been improved no doubt. In the attached, I have marked some remaining errors. There are grammatical, stylistic, and logical problems. The title still has the absurd statement, Future Therapeutic Insights which I would definitely remove. "Neuroinflammation" is a term that is better abandoned (see attached paper in Neuron).

Author Response
Revised version Manuscript n. cancers-2298054
MESSINA, 02nd May 2023
Estimed Guest Editor: Prof. Dr. Gabriella D’Orazi, Dear Reviewers, we send You the newly revised version of our manuscript after having done the changes requested by reviewers 2 and 4 (in yellow ink).
Reviewer 2
We have removed and/or changed all grammatical mistakes underlined by red ink from the reviewer. In addition we have modified the title of the manuscript as requested.
Reviewer 4 Report
The manuscript has been improved significantly.
Minor points:
1. “In particular, it has been shown that cannabinoids as well as antidepressant, may induce cell death of cancer cells by autophagic activation, although non-transformed astrocytes appear resistant to cannabinoid killing action[146,147].”
It should be noted that a combination of an a tricyclic antidepressant (imipramine) and an anticoagulant (ticplopidine) was used in the study from the Hanahan group. Since this is not a general action of this drug classes, but rather a special feature of these particular (repurposed) drugs, please indicate the names of them in the text of the manuscript.
2. Informations in Table 1 need to be made more precise and “mechanism of action” should be restricted to effects related to autophagy. For example in the case of Bortezomib only unrelated effects are indicated.
ABT737 does reduce Bcl-2, but it activates autophagy
3. Please carefully re-check whether all chosen references are appropriate.
“Recently, some reports have documented an hyper-activation of CMA in GBM 204 through the expression of LC3B, LAMP1 and LAMP2A with their downregulation by curcumin [24–26];”
These papers do not refer to hyperactivation of autophagy or to brain tumors. Other work would perhaps be more fitting for this discussion. PMID: 35131870; PMID: 35468038
Author Response
Revised version Manuscript n. cancers-2298054
MESSINA, 02nd May 2023
Estimed Guest Editor: Prof. Dr. Gabriella D’Orazi, Dear Reviewers, we send You the newly revised version of our manuscript after having done the changes requested by reviewers 2 and 4 (in yellow ink).
Reviewer 4
- We have added the specific combination of drugs (antidepressant and anticoagulant) with cannabinoids
- We have modified the Table 1 as requested
- We have checked the references and we have introduced the suggested papers (PMID: 35131870; PMID: 35468038)